

# *Metasychis varicollaris* sp. nov. and report of *Metasychis gotoi* (Maldanidae, Annelida) from the China Seas

Yueyun Wang[1], Xinzheng Li[2,3,4,5] and Chunsheng Wang[1,6]

[1] Key Laboratory of Marine Ecosystem Dynamics, Second Institute of Oceanography, Ministry of Natural Resources, Hangzhou, China
[2] Institute of Oceanology, Chinese Academy of Sciences, Qingdao, China
[3] Center for Ocean Mega-Science, Chinese Academy of Sciences, Qingdao, China
[4] University of Chinese Academy of Sciences, Beijing, China
[5] Laboratory for Marine Biology and Biotechnology, Pilot National Laboratory for Marine Science and Technology (Qingdao), Qingdao, China
[6] School of Oceanography, Shanghai Jiao Tong University, Shanghai, China

## ABSTRACT

Polychaete species are widely distributed throughout Indo-Pacific and European waters. We collected *Metasychis* specimens from the China Seas to report on *Metasychis varicollaris* sp. n. and *Metasychis gotoi* (*Izuka, 1902*) in greater detail. Geographic analysis of the potential distribution areas of *M. gotoi* indicates that it may be found in most coastal areas of China. The newly discovered species, *M. varicollaris* and *M. gotoi*, have an overlapping distribution in the northern South China Sea. *Metasychis varicollaris* sp. n. is characterized by a crenulated cephalic rim, complete collar on chaetiger 1, a packet-shaped anal funnel, and a spirally-fringed notochaetae with spiral pectinate bands imbricated over the main shaft. Our study provides a taxonomic key to all species of *Metasychis*.

## INTRODUCTION

Maldanids, with their segmental bodies, are an easily recognizable polychaete family. Individuals have elongated segments from the median to the posterior regions of the body, with the parapodia resembling slender bamboo-shoots at one end (*Fauchald, 1977*). Maldanids are found in hard or soft substrates from the intertidal region to the deep sea (*Paterson et al., 2009*; *De Assis & Christoffersen, 2011*). Malmgren erected the family Maldanidae in 1867. *Arwidsson (1906)* subsequently divided the family into five subfamilies: Euclymeninae, Lumbriclymeninae, Maldaninae, Nicomachinae and Rhodininae. Three additional subfamilies have since been proposed: Clymenurinae (*Imajima & Shiraki, 1982a*), Bogueinae (*Wolf, 1983*), and Notoproctinae (*Detinova, 1985*). *De Assis & Christoffersen (2011)* proposed the phylogenetic relationships of Maldanidae subgroups based morphological data, however, the subfamilies Clymenurinae and Bogueinae were not supported by the character-based phylogenetic tree estimated using maximum parsimony. Therefore, Clymenurinae was included with Euclymeninae, and Bogueinae with Rhodininae. *Kobayashi et al. (2018)* reconstructed the molecular

Corresponding author
Chunsheng Wang,
wangsio@sio.org.cn

phylogeny and confirmed the monophyly of the subfamilies Rhodininae, Maldaninae, Lumbriclymeninae and Nicomachinae. The subfamily Euclymeninae was shown as monophyletic (*De Assis & Christoffersen, 2011*), but was recovered as paraphyletic and Nicomachinae was clustered within it (*Kobayashi et al., 2018*).

The Maldaninae genus *Metasychis*, was erected by *Light (1991)* to include four species: *M. collariceps* (*Augener, 1906*), *M. disparidentatus* (*Moore, 1904*), *M. fimbriatus* (*Treadwell, 1934*) and *M. gotoi* (*Izuka, 1902*). The members of *Metasychis* are distinguished by their well-developed cephalic rim with crenulations or cirri, J- or U- shaped nuchal grooves, chaetiger 1 with reduced or complete collar, notochaetae on the middle body with spirally fringed distal ends, and a funnel-like pocket anal plate. Only one *Metasychis* species, *Metasychis gotoi*, was recorded from the China Seas (*Liu, 2008*; *Yang & Sun, 1988*) which are located in the western North Pacific and include the Bohai Sea, Yellow Sea, East China Sea and South China Sea. We examined the Maldaninae specimens deposited in the Marine Biological Museum of the Chinese Academy of Sciences (MBMCAS) and describe a new species of *Metasychis* from the northern South China Sea where the species are known to overlap.

## MATERIALS AND METHODS

We examined all of the Maldaninae specimens deposited in the Marine Biological Museum of the Chinese Academy of Sciences (MBMCAS) in the Institute of Oceanology (IOCAS) that were collected during the National Comprehensive Oceanography Survey (NCOS, 1958–1960) and the Sino-Vietnam Joint Comprehensive Oceanographic Survey of Beibu Gulf (1959–1961). The specimens were preserved in a solution of 75% ethanol. The sampling sites are shown in Fig. 1.

The potential geographic distributions of *Metasychis gotoi* were predicted using the MaxEnt program (*Steven, Dudík & Schapire, 2019*) with dismo packages (*Hijmans et al., 2017*) in an R environment. Ten environmental variables (mean of chlorophyll, dissolved oxygen, iron, nitrate, phosphate, phytoplankton, primary productivity, salinity, silicate and temperature at present benthic mean depth) were downloaded from Bio-ORACLE (*Tyberghein et al., 2012*; *Assis et al., 2018*) and 115 presence localities were used in the analysis. Twenty-five percent of the locations were selected randomly for modeling and were evaluated using the evaluate function in dismo package.

We made morphological observations with a Zeiss Stemi 2,000-C stereo microscope and compound microscope. Line drawings were made using a UGEE electronic drawing tablet in Adobe Photoshop. We rinsed the samples for viewing with a scanning electron microscope (SEM) with distilled waters for 12 h to dissolve mineral crystals. We then ran the samples through a series of ethanol concentrations and stored them in absolute alcohol until observations were made.

### Nomenclatural acts

The electronic version of this article in Portable Document Format (PDF) will represent a published work according to the International Commission on Zoological Nomenclature (ICZN), and hence the new names contained in the electronic version are effectively

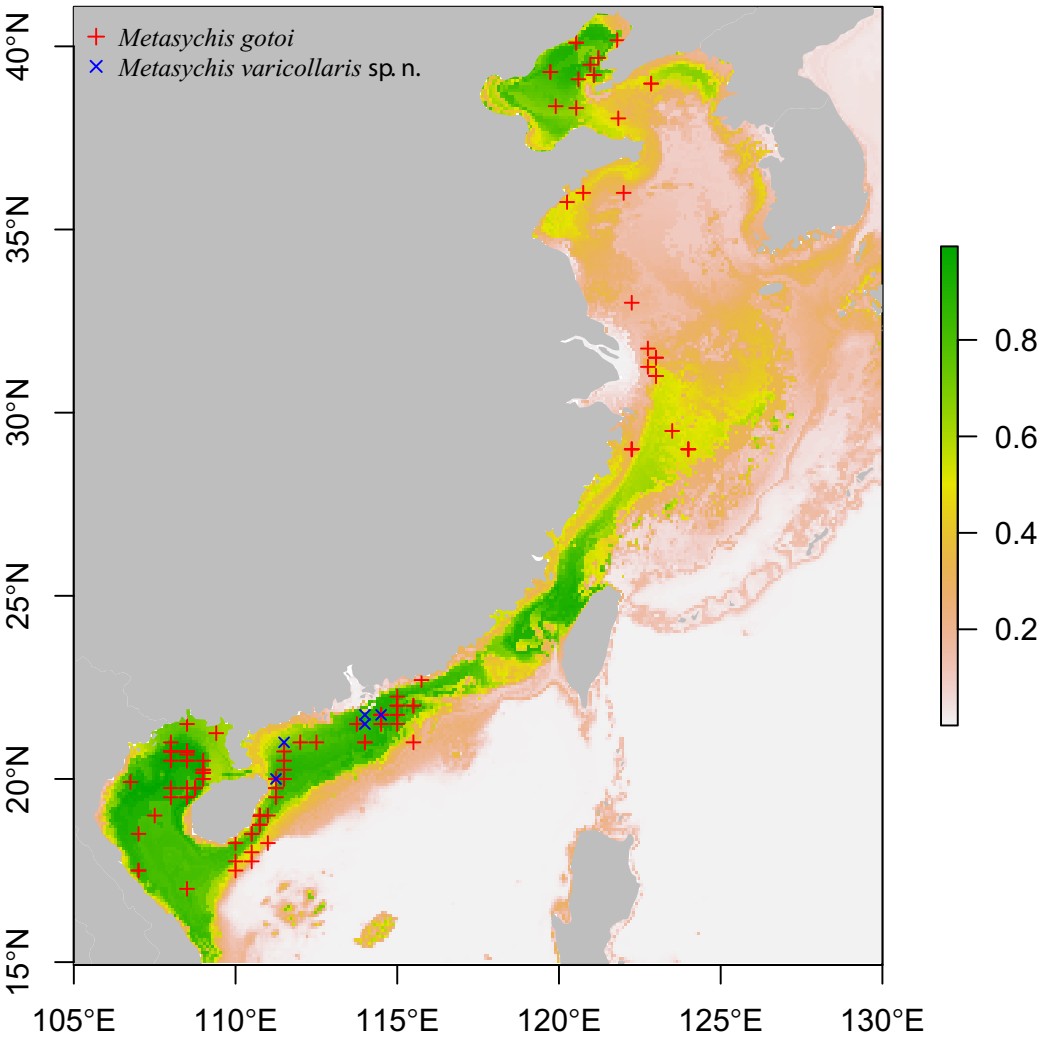

**Figure 1 Sampling sites of *Metasychis varicollaris* sp. n. (×) and *Metasychis gotoi* (+).** Colors indicating predicted probability of suitable conditions for *M. gotoi*.

published under that Code from the electronic edition alone. This published work and the nomenclatural acts it contains have been registered in ZooBank, the online registration system for the ICZN. The ZooBank Life Science Identifiers (LSIDs) can be resolved and the associated information viewed through any standard web browser by appending the LSID to the prefix http://zoobank.org/. The LSID for this publication is: (urn:lsid: zoobank.org:pub:A018F8D0-F9A6-4D64-B206-4FFB39160032). The online version of this work is archived and available from the following digital repositories: PeerJ, PubMed Central and CLOCKSS.

# RESULTS
Family Maldanidae *Malmgren, 1867*
Subfamily Maldaninae *Malmgren, 1867*

Genus *Metasychis Light, 1991*
*Metasychis Light, 1991*: 133–146; *Wang & Li, 2016*: 13.

Type species: *Metasychis disparidentatus* (*Moore, 1904*)

**Diagnosis** (after *Light, 1991*, different feature highlighted in italicized). Body with 19 chaetigers, without neurochaetae on the first chaetigers. Lateral cephalic rim with crenulations or digitate cirri, fusing with expanded prostomial palpode or setting off from it by furrows, *connecting to J- or U-shaped nuchal groove or not connecting to nuchal groove*. Collar on chaetiger 1 complete, or reduced to a thick ventral roll of tissue. Notochaetae including spirally-fringed fimbriae. One pygidial achaetigerous segments or none. No anal valve. Pygidium well developed, forming a deep, posterior, funnel-like pocket, with a pair of deep lateral notches. Dorsal lobe of the pygidium with or without cirri.

**Remarks**. In *Light's (1991)* description, the *Metasychis* species usually has type B notochaetae, in which the fimbriae are more delicate and expanded away from the shaft (sometimes type A) in which the fimbriae are spinose and closely imbricated over the main shaft. The notochaetae examined here in *M. varicollaris* sp. n. and *M. gotoi* are closer to type A notochaetae in *Light (1991)*.

Several specimens with a distinct collar were observed in the *Metasychis* material in the Marine Biological Museum of the Chinese Academy of Sciences and they should belong to a new species. They are described below.

### *Metasychis varicollaris* sp. n.
(Figs. 2 and 3)

**Material examined**. Holotype. MBM 012597, South China Sea, st. 6052, 21.5°N, 114°E, 54.5 m depth, 9 April 1959. Complete specimen, length ca. 67 mm, width ca. 2.2 mm at chaetiger 1, with muddy tube encompassment. Paratypes. MBM 012647, South China Sea, st. 6045, 21.75°N, 114.5°E, 61 m depth, 20 March 1959. Anterior fragment with 10 chaetigers. Chaetigers 11– 12 were used in SEM examination. MBM 012658, South China Sea, st. 6045, 21.75°N, 114.5°E, 59.6 m, muddy sediment, 8 April 1960. MBM 012676, South China Sea, st. 6116, 21°N, 111.5°E, 41 m depth, muddy sediment, 12 April 1959. Other specimens examined. MBM 012576, South China Sea, st. 6051, 21.75°N, 114°E, 44 m, muddy sediment, 9 December 1959. MBM 012674, South China Sea, st. 6131, 20°N, 111.25°E, 50 m, muddy sediment, 6 April 1960. MBM 012645, South China Sea, st. 6131, 20°N, 111.25°E, 44 m, 29 October 1959.

**Description**. Body cylindrical, with 19 chaetigers, and a funnel-shaped pygidium (Figs. 2A–2D, 3E and 3F). Body color in alcohol yellow. The first 6–7 parapodial tori with glandular pads (Fig. 3A). Anterior end obliquely truncate, with an elliptical cephalic plate (Figs. 2B, 2E and 3D). Cephalic rim divided into three parts by a pair of deep lateral notches. Triangular to rounded crenulations on cephalic rim well-developed; 4–6 crenulations on lateral part, 12–16 on posterior part (Figs. 2B and 3D). Prostomial palpode

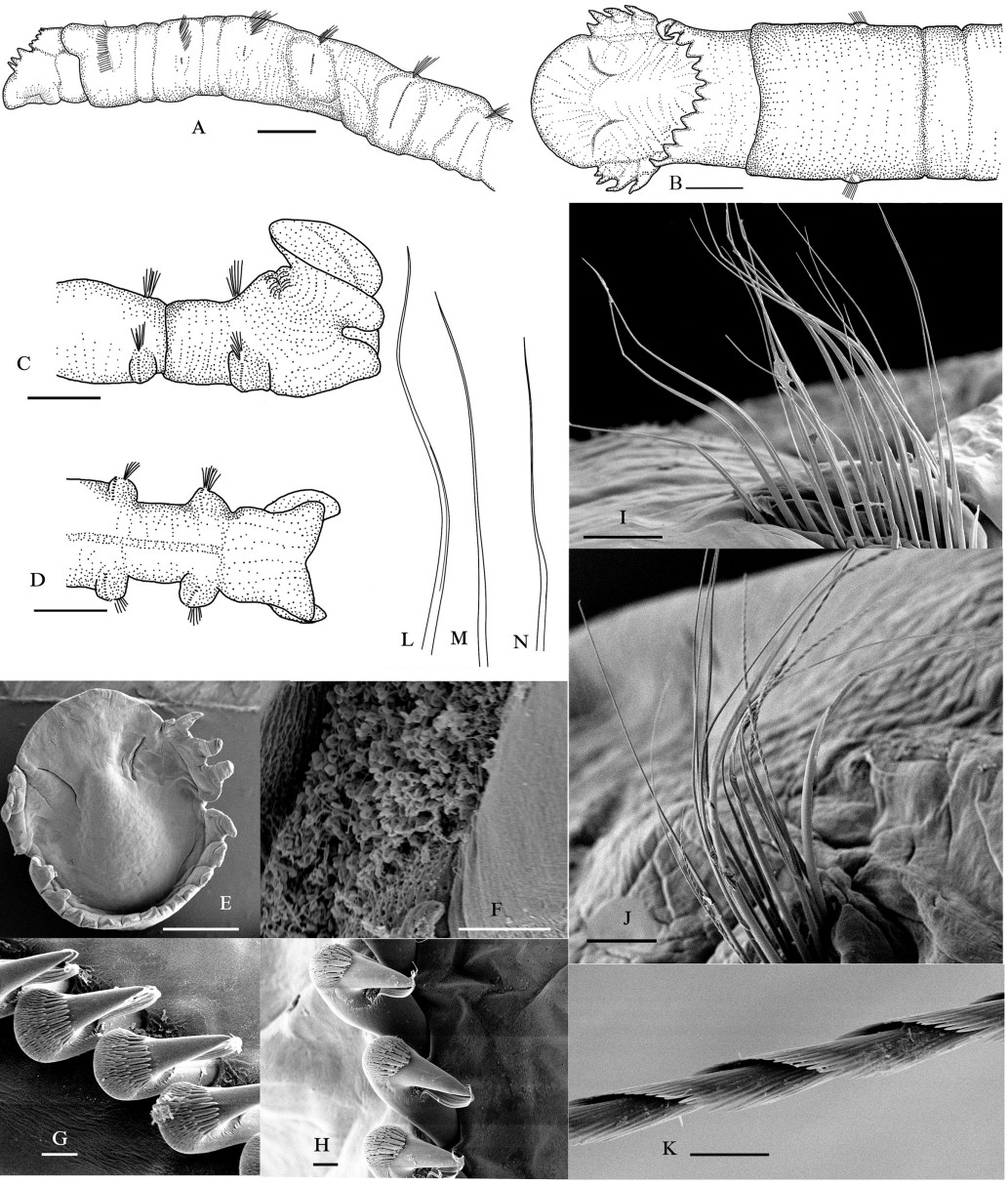

**Figure 2** *Metasychis varicollaris* **sp. n.** (A) Anterior region in lateral view. (B) Cephalic plate in dorsal view. (C) Pygidium in lateral view. (D) Pygidium in ventral view. (E) Cephalic plate in dorsal view. (F) Nuchal groove. (G) Neurochaetae in chaetiger 6. (H) Neurochaetae in chaetiger 11. (I) Notochaetae in chaetiger 5. (J) Notochaetae in chaetiger 11. (K) Spinose part of notochatae. (L) Limbate capillary. (M) Common capillary. (N) Geniculate notochaetae. Scale bars: 1.0 mm (A–E), 10 μm (F–H and K), 100 μm (I and J).

broadly rounded. Eyes absent. Nuchal groove curved, slightly J-shaped (Figs. 2B and 2E), with many small curly cilia (Fig. 2F). Cephalic keel remarkable, high and long, wider posteriorly (Figs. 2B and 2E).

First three chaetigers relatively short, about 1–2 times as long as wide, biannulate in lateral view (Figs. 2A and 3A). Prominent complete collar on chaetiger 1. Dorsal part well-developed, longer than ventral part, extending forward (Figs. 2A, 3B and 3C).

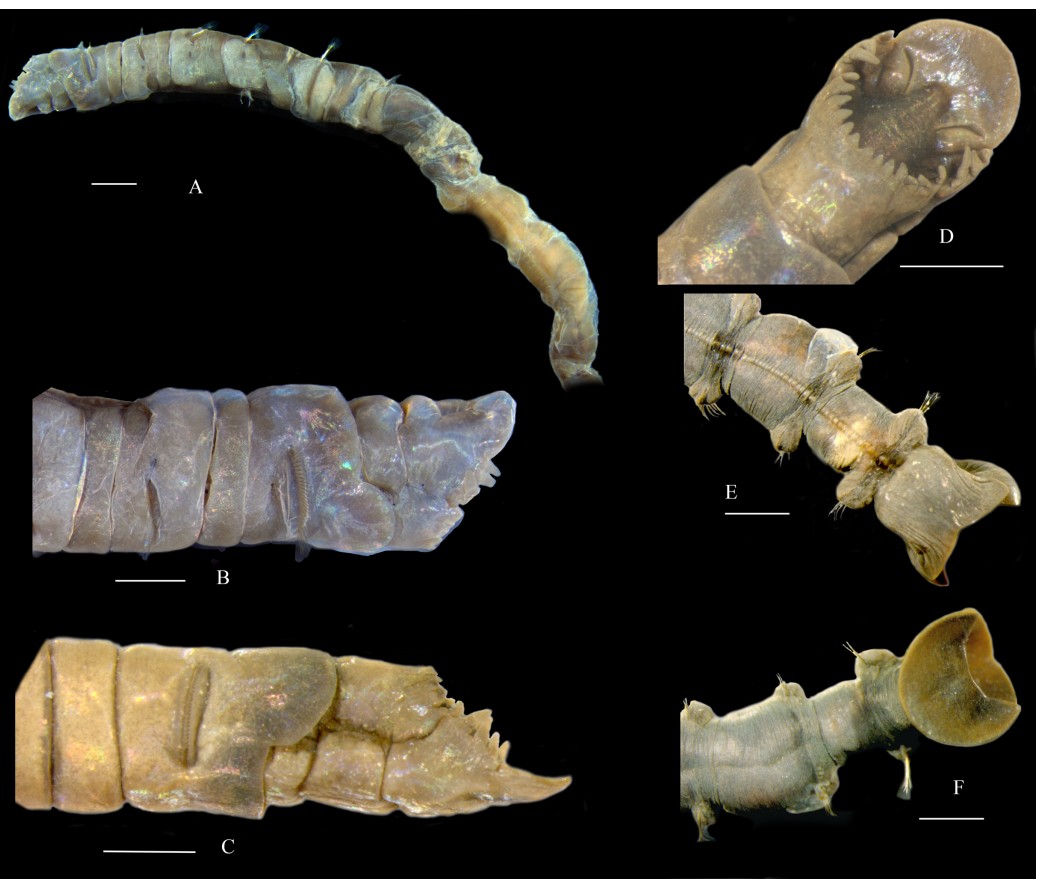

**Figure 3** *Metasychis varicollaris* **sp. n.** (A) Anterior region in lateral view. (B and C) Head in lateral view. (D) Head in dorsal view. (E) Pygidium in ventral view. (F) Pygidium in dorsal view. Scale bars: 1.0 mm.

Mid-body, and posterior chaetigers typically with inflated neuropodial tori. Neurochaetae present from chaetiger 2, typically rostrate uncini similar on all chaetigers without significant variation between the first three uncini from subsequent uncini, arranged in a row on neuropodial tori (Figs. 2G and 2H). Capitium of uncinus with 5–6 transverse arcs of small teeth. First arc with about 12 small teeth larger than on other arcs. A tuft of bristles under main fang. Anterior chaetigers with two kinds simple capillary notochaetae (Fig. 2I): limbate capillary with narrow wing on one side (Fig. 2L) and common capillary without similar structures (Fig. 2M). Middle and posterior chaetigers with long spirally-fringed notochaetae and companion geniculate notochaetae (Figs. 2J, 2K and 2N). Long spirally-fringed notochaetae with two spirally pectinate bands imbricated over the main shaft.

Pre-pygidial achaetigerous segment absent. Anal mound well-developed (Figs. 2C, 3E and 3F). Anal pore without anal valve. Anal funnel elliptical in end view. Deep lateral notches separating anal funnel into dorsal and ventral lobes. Dorsal lobe expanded, disc-shaped, without marginal cirri observed. Ventral lobe forming shallow posterior pocket, with a widen midventral notch.

**Etymology**. "vario", Latin: different, various; "collare", Latin: collar, neck. The specific name *varicollaris* referres to the collar shape of this species different from that of congeneric members.

**Distribution**. Northern South China Sea.

**Remarks**. *Metasychis varicollaris* sp. n. is morphologically similar to *M. gotoi*, especially in body size and cephalic plate. However, the new species has a fully developed collar in chaetiger 1, as opposed to a ventral collar in *M. gotoi*. *Metasychis collariceps* (*Augener, 1906*) and *M. fimbriatus* (*Treadwell, 1934*) also have a complete collar on chaetiger 1. The new species can be distinguished from the two species by the shape of collar and cephalic rim. Collar is laterally notched in *M. collariceps* but is full in the new species. The margin of the posterior cephalic rim is complete in *M. fimbriatus* but is crenulated in the new species.

### *Metasychis gotoi* (*Izuka, 1902*)
(Fig. 4)

*Maldane gotoi Izuka, 1902*, p. 109, Pl. 28, figs. 1–8
*Asychis gotoi* (*Izuka, 1902*)—*Imajima & Shiraki, 1982b*, p. 75, figs. 36a–l; *Yang & Sun, 1988*, pp. 264–265, figs. 125F–K
*Maldane coronata Moore, 1903*, p. 483–485, Pl. 28, figs. 94–96
*Metasychis gotoi* (*Izuka, 1902*)—Light, 1991, figs.1L–M

**Material examined**. MBM 006305–006307; 006310–006312; 006317; 006320; 006347; 006355; 006412; 007966; 007967; 008113; 008119; 008138; 012498; 012518; 012564–012566; 012569; 012571; 012573–012574; 012577–012580; 012582; 012586; 012588–012591; 012593;012603–012607; 012611; 012615–012619; 012621–012626; 012628; 012630; 012633; 012636; 012640–012643; 012646; 012648; 012650–012652; 012654–012655; 012657; 012660; 012664–012665; 012668–012670; 012675; 012677; 012679; 012681; 012685–012687; 012708; 012715; 012730; 201449–201455; 201457–201461; 201463; 201466; 201475–201492. Speciemens were collected from the Bohai Sea, Yellow Sea, East China Sea and northern South China Sea. Location information provided in Supplemental Files 3.

**Diagnosis**. Cylindrical body with nineteen chaetigers. Chaetiger 1 with a short ventral collar (Figs. 4B and 4C). First four chaetigers biannulate dorsally, and usually with epidermal glands. Following 5–6 chaetigers only with ventral epidermal glands (Fig. 4D).

Cephalic plate elliptical (Fig. 4A). Prostomial palpode broadly rounded, mushroom-shaped. Cephalic rim developed, divided into three parts by two lateral notches. Lateral cephalic rim with 5–7 digitate cirri (Figs. 4A–4C). Posterior rim with irregular crenulations, sometimes with several small cirri. Cephalic keel short and broad. Nuchal groove curved, slightly J–shaped, extending outwards and forwards, forming a faint notch separating lateral cephalic rims from prostomial palpode.

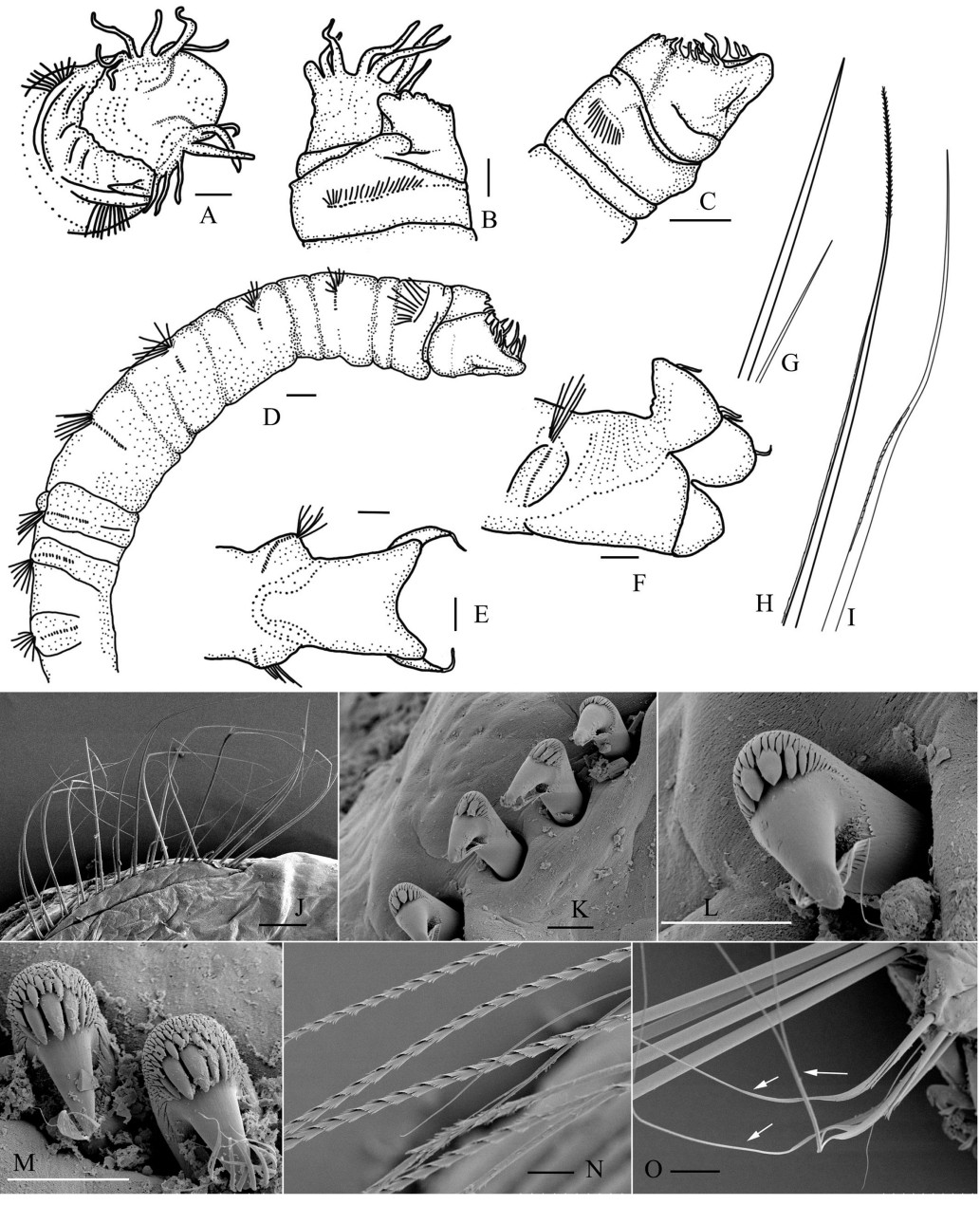

**Figure 4** *Metasychis gotoi* (*Izuka, 1902*). (A) Head region in dorsal view. (B and C) Head region in lateral and ventral views, respectively. (D) Anterior segments in lateral view. (E and F) Pygidium in ventral and lateral view. (G) Capillary notochaeta and short slender companion notochaetae on anterior segments. (H and I) Limbate notochaeta with spirally fringed tip and geniculate notochaeta on middle segments. (J–O) SEM images of chaetae. (J) Notochaetae on chaetiger 2. (K and L) Neurochaetae on chaetiger 2. (M) Uncini on chaetiger 11 in apex view. (N) Spirally fringed notochaetae. (O) Geniculate companion notochaetae. Scale bars: 0.5 mm (A–F), 250 μm (J), 20 μm (K–N), 50 μm (O).

Anal plate well developed, divided into a flaring dorsal lobe and a deep funnel-shaped ventral lobe (Figs. 4E and 4F). Margin of dorsal lobe usually with six slender cirri.

Notochaetae arranged in two rows. Anterior chaetigers with simple capillary notochaetae including stout notochaetae and short companion chaetae (Figs. 4G and 4J). Middle and posterior chaetigers with geniculate companion chaetae and spirally fringed notochaetae, spinose spiral bands closely imbricated over main shaft (Figs. 4H, 4I, 4N and 4O). Chaetiger 1 without neurochaetae. Neurochaetae from chaetiger 2, rostrate uncini with several transversal rows of small teeth on main fang (Figs. 4K–4M).

Tube encrusted with mud.

**Distribution**. *Metasychis gotoi* is widely distributed in the Indo-Pacific Ocean (*Fauvel, 1932*; *Yang & Sun, 1988*; *Liu, 2008*) and may be introduced in the Mediterranean Sea (*Zenetos et al., 2010*). Predicted potential distribution shows that *M. gotoi* may occur in most coastal areas of China (Fig. 1).

**Remarks.** *Metasychis gotoi* is distinguishable from other species of *Metasychis* by its developed crenulated cephalic rim, ventral collar on chaetiger 1 and anal cirri. This combination of characteristics is most similar to *M. disparidentatus*. However, there is no cirrus on the anal plate of *M. disparidentatus*.

## DISCUSSION

*Light (1991)* revised the subfamily Maldaninae and listed four *Metasychis* species (*M. collariceps*; *M. disparidentatus*; *M. fimbriatus*; *M. gotoi*). *Wang & Li (2016)* proposed a key to distinguish the Maldaninae genera. Based on those definitions, *Asychis* has no collar on chaetiger 1. Three genera of Maldaninae, *Chirimia* (*Light, 1991*), *Metasychis* (*Light, 1991*), and *Sabaco* (*Kinberg, 1867*) have a collar on chaetiger 1; *Sabaco* is characterized by crescentic nuchal grooves and a smooth cephalic rim. *Chirimia* and *Metasychis* have a mushroom-shaped palpod, and J or U-shaped nuchal grooves. *Chirimia* is distinguishable from *Metasychis* by the presence of an anal valve. Additionally, the pygidium of *Metasychis* is more developed than that of *Chirimia*. *Metasychis collariceps* was first described as a member of genus *Maldane* (*Augener, 1906*). *Hartman (1938)* transferred it to the genus *Asychis*. *Light (1991)* revised the subfamily Maldaninae and transferred it to the genus *Metasychis*. *Metasychis collariceps* has a complete collar on chaetiger 1 and a dentate lateral cephalic rim, based on its original description. Additional information is needed to confirm its taxonomic status. The species identification has been temporarily assigned based on the information that was available at the time. *Metasychis disparidentatus* is type species of this genus designated by *Light (1991)*. It has a collar limited to the ventral side of chaetiger 1, J-shaped nuchal grooves, and a well-developed pygidium. *Metasychis fimbriatus* was first described as a member of genus *Maldanella* by *Treadwell (1934)*. *Hartman (1956)* transferred it to the genus *Asychis*. Later, *Light (1991)* transferred it to the genus *Metasychis*. It has a complete collar on chaetiger 1 and a well-developed pygidium with cirri on its dorsal lobe based on original description (*Treadwell, 1934*).

## CONCLUSIONS

Maldaninae is a poorly known subfamily of Maldanidae because of inadequate descriptions of early-described species, requirements for complete specimens for complete

identification. Correct taxonomy is critical for biodiversity mapping and environmental surveillance monitoring. The present study reported the most comprehensive survey of *Metasyshis* species from coastal waters of China, detailed information of taxonomy and distribution. The description of new *Metasyshis* species from southern China contributes to better understand its diversity worldwide. To date, members of *Metasychis* are reported to have limited geographical distribution except *M. gotoi*. *Metasychis collariceps* distributed in Caribbean Sea, *M. disparidentatus* from western Canada south to Southern California and Japan, *M. fimbriatus* is distributed in Puerto Rico. The five species may be distinguished by the following key:

### Key to species of *Metasychis* Light, 1991

1. Fully developed collar on chaetiger 1 ............................................................................2
   Collar limited to ventral side of chaetiger 1 ..............................................................4
2. Collar entire without lateral notches .............................................................................3
   Collar with lateral notches..................................................**M. collariceps** (*Augener, 1906*)
3. Posterior part of cephalic rim crenulated....................................**M. varicollaris** **sp. nov.**
   Posterior part of cephalic rim entire ..............................**M. fimbriatus** (*Treadwell, 1934*)
4. Cephalic rim with faint crenulations; anal plate without cirri ............................................
   .............................................................................**M. disparidentatus** (*Moore, 1904*)
   Lateral lobes of cephalic rim usually with digitate cirri; dorsal lobe of the anal plate with slender cirri ..................................................................**M. gotoi** (*Izuka, 1902*)

## ACKNOWLEDGEMENTS

The authors with to thank VLIZ Library for providing important references as well as Dr. De Assis and two anonymous reviewers for giving us valuable comments and suggestions on the manuscript. We also thank the managers of the Marine Biological Museum of the Chinese Academy of Sciences for their help in sorting the material.

### Funding

The research was funded by the National Natural Science Foundation of China (No. 41806179) and the Ocean Public Welfare Scientific Research Project, State Oceanic Administration of the People's Republic of China (No. 201505004). The funders had no role in study design, data collection and analysis, decision to publish, or preparation of the manuscript.

### Grant Disclosures

The following grant information was disclosed by the authors:
National Natural Science Foundation of China: 41806179.
Ocean Public Welfare Scientific Research Project, State Oceanic Administration of the People's Republic of China: 201505004.

## Competing Interests

The authors declare that they have no competing interests.

## Author Contributions

- Yueyun Wang conceived and designed the experiments, performed the experiments, analyzed the data, prepared figures and/or tables, authored or reviewed drafts of the paper, and approved the final draft.
- Xinzheng Li conceived and designed the experiments, authored or reviewed drafts of the paper, and approved the final draft.
- Chunsheng Wang conceived and designed the experiments, authored or reviewed drafts of the paper, and approved the final draft.

## Data Availability

Environmental data is available at Figshare:

Yueyun (2020): Ten env-asc layers.zip. figshare. Dataset. DOI 10.6084/m9.figshare.12560171.v1

Code and raw data are available in the Supplemental Files.

All specimens described in the manuscript are stored in the Marine Biological Museum of the Chinese Academy of Sciences (MBMCAS) in the Institute of Oceanology (IOCAS).

## New Species Registration

The following information was supplied regarding the registration of a newly described species:

Publication LSID: urn:lsid:zoobank.org:pub:A018F8D0-F9A6-4D64-B206-4FFB39160032.

*Metasychis* LSID: urn:lsid:zoobank.org:act:9607E46E-D7BE-4B24-A3A3-1F107A2BF333.

*Metasychis varicollaris* LSID: urn:lsid:zoobank.org:act:0EDF3656-40CC-4F35-BFCE-DE3E76EEA09B.

## Supplemental Information

Supplemental information for this article can be found online at http://dx.doi.org/10.7717/peerj.10608#supplemental-information.

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
