# Peer review of "Metasychis varicollaris sp. nov. and report of Metasychis gotoi (Maldanidae, Annelida) from the China Seas"

_PeerJ, doi:10.7717/peerj.10608_

## Round 0.1 · original submission · Major Revisions

I have heard back from three reviewers, who were all positive about your work, and have all provided helpful comments to help make your work better. Please consider their comments carefully, and I look forward to seeing a revised version of your work.

·

Basic reporting

Dear Editor of Pee Journal Dr. James Reimer

I would like to thanks to authors for suggest my name to review this paper, especially be a group of Maldanidae, which worked for almost seven years.

I would like to give some suggestions on manuscript entitled: Metasychis varicollares sp. nov., and report of Metasychis gotoi (Maldanidae, Annelida) from China Seas by Yueyun Wang, Xinzheng Li, Chunsheng Wang.

In general this manuscript reasonable, and deserve to be published after many changes. I didn’t understand some sentences and expressions, principally in relation to the description of the new species, but I pointed on the manuscript. Please, if the author need some informations I’ll be always to dispostion. I’m not a native speaker English, but according to my experience in the family Maldanidae, and in this genus, I was thoroug. I think that Dr G.W. Rouse and/or Dr. James Blake will make a better review of the text in English, and improve it in general. THE GRAMMAR OF THE ENGLSHI LANGUAGE NEED TO BE GREATLY IMPROVED.

TITLE: I made few changes in the title, and suggestion.
ABSTRACT: Need improved.
INTRODCTION: Need improved.
MATERIAL AND METHODS:It’s good.
SYSTEMATICS: Improved the ms in general.
REFERENCES: Plese, check citation and references in general. They need to improved

ALL SUGGESTIONS ARE IN THE MSN

Experimental design

No comment

Validity of the findings

No comment

Additional comments

I would like to thanks to authors for suggest my name to review this paper, especially be a group of Maldanidae, which worked for almost seven years.
I would like to give some suggestions on manuscript entitled: Metasychis varicollares sp. nov., and report of Metasychis gotoi (Maldanidae, Annelida) from China Seas by Yueyun Wang, Xinzheng Li, Chunsheng Wang.
The manuscript need substancial changes.

Reviewer 2 ·

Basic reporting

This paper is reported a new species of Metasychis and M. gotoi from China Seas. The new species is well described. The report represents a contribution that further improves the knowledge of the biodiversity of this group and of polychaetes.

I recommend some minor modifications for improving the manuscript.
1. Please provide photographs of the specimen like Wang et al. (2016) Fig. 3.
2. Fig. 2B should be revised because it does not contain the anterior margin of chaetiger 1.
3. I think that M. disparidentata sensu Imajima and Shiraki (1982) also resembles the new species. A comparison with Imajima's descriptions will provide a better understanding of the relationships between the Asian Metasychis species.

Additional comments are provided in the PDF file.

Yours sincerely

Experimental design

no comment

Validity of the findings

no comment

Annotated reviews are not available for download in order to protect the identity of reviewers who chose to remain anonymous.

Reviewer 3 ·

Basic reporting

The manuscript describes a new species of marine worm from China and reports on an additional maldanid species that is widely distributed in China seas. The authors describe the new species using the appropriate techniques (Light and electron microscopy) and there is no question that the proposed new species is indeed undescribed.

The introduction is missing some recent literature about the phylogeny of maldanids and I have indicated in the revision.

The description needs to be improved to follow the telegraphic style and I have provided many suggestions throughout the manuscript.

The figures are of good quality but the authors should re-think about ordering them to follow a more standard structure (A to C, left to right, for example).

Experimental design

The manuscript is within the scope of PeerJ in the Biological Sciences field, although it is only reporting on two species (one newly described) with limited impact. The authors did not include any molecular data but as a taxonomist myself, I think that it should not be discourage publication hence morphological description is well performed.

The methods section is well described and with sufficient detail.

Validity of the findings

The paper describes a new polychaete species so the findings are novel per se.

Additional comments

I have provided many suggestions to improve the flow of the manuscript, especially the descriptions that should be in telegraphic style. Additional comments and suggestions are provided in the attached PDF file.

Annotated reviews are not available for download in order to protect the identity of reviewers who chose to remain anonymous.

---

## Round 0.2 · Minor Revisions

I have heard back again from all three reviewers, who are much happier with this current version, and also offer some additional minor edits. As well, I believe the title and some figure legends need some consideration, as "the China Seas" does not seem to be a common term, and as well your specimens include many from Vietnam, yet no country names are given in some of your specimen lists; thus "the China Seas" seems a bit innaccurate. Please look at the reviewers' helpful comments, and I look forward to seeing your revised work soon.

·

Basic reporting

Dear Editor
I want to thank you and authors

The authors did all substantial changes pointed the anterior review. I make many happy with all changes. Although, small errors persisted. For example: In main text we can find: citation of two kinds: Author & Author - Author and Author. Please, the authros can be consistent. There are some erros in references, for example: After name of journal, we can see with can or point, or nothing. Please, can be consistent. Some references are missing Volume.
The added the discussion and it is good now.
In general, the ms is good and can be published after minor errors.
Thank you very much

Experimental design

Dear Editor
I want to thank you and authors

The authors did all substantial changes pointed the anterior review. I make many happy with all changes. Although, small errors persisted. For example: In main text we can find: citation of two kinds: Author & Author - Author and Author. Please, the authros can be consistent. There are some erros in references, for example: After name of journal, we can see with can or point, or nothing. Please, can be consistent. Some references are missing Volume.
The added the discussion and it is good now.
In general, the ms is good and can be published after minor errors.
Thank you very much

Validity of the findings

Ok

Additional comments

Dear Editor
I want to thank you and authors

The authors did all substantial changes pointed the anterior review. I make many happy with all changes. Although, small errors persisted. For example: In main text we can find: citation of two kinds: Author & Author - Author and Author. Please, the authros can be consistent. There are some erros in references, for example: After name of journal, we can see with can or point, or nothing. Please, can be consistent. Some references are missing Volume.
The added the discussion and it is good now.
In general, the ms is good and can be published after minor errors.
Thank you very much

Reviewer 2 ·

Basic reporting

This version of the manuscript is much improved. Before accepting this manuscript, I suggest some modifications. I have attached the pdf with comments.
If the changes are followed as suggested, I do not need to revise the manuscript again.

Experimental design

no comment

Validity of the findings

no comment

Annotated reviews are not available for download in order to protect the identity of reviewers who chose to remain anonymous.

Reviewer 3 ·

Basic reporting

The manuscript is sound and describes a new species and record of maldanid polychaete from China. I have included additional edits and suggestions in the attached PDF file. The authors have attended my previous suggestions. The descriptions are appropriate and there are no questions about the validity of the proposed species. Other than that, the manuscript meets the general standards of PeerJ and there are no obvious concerns regarding acceptability of the results, completeness of the data provided and the conclusions made.

Experimental design

N/A

Validity of the findings

No additional comment

Additional comments

Please see attached PDF document with minor suggestions to improve the text. As I have commented in the manuscript, I suggest the authors to contact Dr. Geoff Read and make the appropriate update in the WoRMS database before publication of the manuscript. There is no need to state in a published manuscript that a dynamic database is outdated.

Annotated reviews are not available for download in order to protect the identity of reviewers who chose to remain anonymous.

---

## Round 0.3 · Minor Revisions

I have gone over your revisions, and they are mostly well done. Thank you for your efforts. There are still some outlying issues, as below, so my decision is again that minor revisions are needed.

There is an error on line 56, "the China Sea" is not correct, which sea do you mean?

As well, and again, the "China Seas" is not a generally accepted term. The references you included in your reubuttal include works from China, and also a work by Schottenhammer, who actually uses quotation marks in the title of their work as this is not a true geographical term, and goes on to remark in the Abstract of the same work:

> Also in order to maintain the spatial concept operable, we have decided to call this maritime space the “China Seas”, a term that may not be misunderstood in the sense that this East Asian body of water as a whole or all of the sections that we address in this paper at any time belonged to China or were part of Chinese sovereignty. Notwithstanding the fact that the focus of this article lies in the importance and role of maritime space for and in China’s history, we will now and again also discuss developments that took place in Japanese or Korean coastal waters.

Thus, to me, to prevent any claims of sovereignty, you need to clearly define the term "China Seas" in your paper, or change the title to "waters of East Asia", which is not unduly long at all. You have also not added country names into your specimen information for specimens, although you state in the rebuttal "Some specimens we examined here indeed collected from Vietnam", which is not best practice for taxonomy, please do so.

---

## Round 0.4 · accepted · Accept

Thank you for the final edits. I am happy to now move this work into production, and look forward to seeing your final published version online. Congratulations!